# Innovative Invasive Loco-Regional Techniques for the Treatment of Lung Cancer

**DOI:** 10.3390/cancers15082244

**Published:** 2023-04-11

**Authors:** Erik Claes, Reinier Wener, Arne P. Neyrinck, Axelle Coppens, Paul E. Van Schil, Annelies Janssens, Thérèse S. Lapperre, Annemiek Snoeckx, Wen Wen, Hanne Voet, Stijn E. Verleden, Jeroen M. H. Hendriks

**Affiliations:** 1ASTARC (Antwerp Surgical Training, Anatomy and Research Centre), University Hospital Antwerp, Drie Eikenstraat 655, 2650 Edegem, Belgium; 2Department of Thoracic and Vascular Surgery, University Hospital Antwerp, Drie Eikenstraat 655, 2650 Edegem, Belgium; 3Department of Pulmonology, University Hospital Antwerp, Drie Eikenstraat 655, 2650 Edegem, Belgium; 4Anesthesia and Algology Unit, Department of Cardiovascular Sciences, KU Leuven, 3000 Leuven, Belgium; 5Department of Thoracic Oncology, University Hospital Antwerp, Drie Eikenstraat 655, 2650 Edegem, Belgium; 6LEMP (Laboratory of Experimental Medicine and Pediatrics), University Antwerp, Universiteitsplein 1, 2610 Wilrijk, Belgium; 7Faculty of Medicine and Health Sciences, University Antwerp, Universiteitsplein 1, 2610 Wilrijk, Belgium; 8Department of Radiology, University Hospital Antwerp, Drie Eikenstraat 655, 2650 Edegem, Belgium

**Keywords:** lung cancer, endobronchial, endovascular, transthoracic, inhalation, regional perfusion

## Abstract

**Simple Summary:**

Every year lung cancer takes the lives of many patients, making it the most common cause of cancer-related deaths. While surgery is the gold standard treatment for early-stage lung cancer, its share in treating more advanced disease stages is limited. Therefore, clinicians advise a multimodal treatment consisting of chemotherapy, radiotherapy and/or immunotherapy to treat advanced lung cancers. There has been a growing interest in loco-regional techniques as they are expected to have advantages over current therapies. In this article, we provide an overview of the established and promising innovative invasive loco-regional techniques ordered by their route of administration (endobronchial, endovascular and transthoracic route) and an overview of their implementation and effectiveness. Although the results of these techniques show improved local disease control and effect, there is a need for more clinical studies to guarantee their efficacy and safety before they can be used within the clinic.

**Abstract:**

Surgical resection is still the standard treatment for early-stage lung cancer. A multimodal treatment consisting of chemotherapy, radiotherapy and/or immunotherapy is advised for more advanced disease stages (stages IIb, III and IV). The role of surgery in these stages is limited to very specific indications. Regional treatment techniques are being introduced at a high speed because of improved technology and their possible advantages over traditional surgery. This review includes an overview of established and promising innovative invasive loco-regional techniques stratified based on the route of administration, including endobronchial, endovascular and transthoracic routes, a discussion of the results for each method, and an overview of their implementation and effectiveness.

## 1. Introduction

Cancer is the most common cause of death, taking the lives of 9.96 million people per year worldwide [1]. In 2020, lung cancer was estimated to have caused 1.80 million deaths, and 2.2 million new cases were recorded, making it the largest group of cancer-related deaths [1]. Lung cancers are divided into two main subtypes, non-small-cell lung cancer (NSCLC) and small-cell lung cancer (SCLC). NSCLC covers 84% of diagnoses, including adenocarcinoma, squamous cell carcinoma and large-cell carcinoma, while SCLC accounts for 13% and other tumor types (carcinoids, sarcomas and lymphomas) for 3% [2].

Positron emission tomography and computed tomography (PET-CT) is an essential diagnostic tool in the management of lung cancers. It combines functional and anatomical information, allowing precise disease staging and thus better decision-making in treatment options. The eighth edition of the tumor, node and metastasis (TNM) classification system, published in 2017, is the globally accepted standard for the staging of lung cancer. Based on this classification system, the European Society for Medical Oncology (ESMO) has proclaimed surgery to be the standard treatment for early-stage lung cancers (stages 0, I and IIa) [3,4,5]. Depending on the type of cancer and tumor size, surgery involves a lobar or sublobar resection. Unfortunately, most cancers get diagnosed in more advanced disease stages, including stages III and IV, for which surgical resection is usually not recommended. The severity of the procedure and the presence of widely spread metastases make surgery unsuitable for advanced cases. Therefore, stage III and IV lung cancer patients receive a multimodal treatment regimen consisting of chemotherapy, radiotherapy and/or immunotherapy [3,4,5]. In addition to these standard treatments, other options include targeted therapy and loco-regional techniques. Targeted therapy blocks the expression of specific molecules associated with cancer growth and progression. It is mainly used to treat NSCLC with mutations in EGFR, ALK, ROS1, BRAF, MET, RET and NTRK genes. There is a growing interest in loco-regional techniques due to their advantages over standard treatments. They are more parenchyma-sparing and are suitable to be performed on patients for whom traditional surgery is not recommended. In addition, they can be used as a treatment modality on their own or as concomitant therapy, making them applicable in the treatment of both early and advanced-stage lung cancers.

Some of these regional administration techniques were discovered decades ago but needed adjustments to fit modern standards or needed to be more effective, while others were developed more recently. This article provides an overview of the established and promising invasive regional treatment techniques based on their administration route, including endobronchial, endovascular and transthoracic routes, a discussion of the results for each method and an overview of their implementation and suitability (Figure 1). Meta-analyses, systematic reviews, (pre-)clinical trials and randomized clinical trials were searched on PubMed until December 2022. An overview of the included articles is given in Appendix A Table A1.

## 2. Endobronchial Administration

### 2.1. Endobronchial Localization and Diagnosis

Exact endobronchial localization of a lesion is essential for endobronchial treatment. Roughly, the airways can be divided into a central and peripheral zone. Central airway lesions can be accessed with rigid bronchoscopy combined with high-energy therapy such as laser, electrocoagulation or cryotherapy. Often, these treatments are not curative and require additional treatment such as radiation therapy or surgery. Conventional bronchoscopy is an essential tool for diagnosing lung lesions with a low risk of complications. However, for peripherally located lung lesions it is not an appropriate technique, and its diagnostic yield is only 18–62% [6,7].

To improve the diagnostic yield of an endobronchial approach, ultrathin bronchoscopes with an outer diameter of 3 mm or less have been developed [8]. In addition, radial endobronchial ultrasound is also available to confirm access to the lesion [8]. However, the structure of the airways is highly variable, and complex branching makes it very difficult to be certain of the location beyond the first few divisions. Therefore, more precise navigation techniques have emerged, such as virtual bronchoscopic navigation (VBN), electromagnetic navigation bronchoscopy (ENB) and robotic guided bronchoscopy. Virtual bronchoscopic navigation is a tool in which 3D reconstructions are generated from a helical chest CT. This creates a virtual roadmap to the lesion of interest. Real-time matching of virtual and actual bronchoscopic images allows a more accurate pathway selection. The diagnostic yield using this approach has been estimated to be up to 74% [9,10,11].

Electromagnetic navigation bronchoscopy uses a technique that involves the generation of an electromagnetic field around the patient. Virtual bronchoscopy images are aligned to the patient’s airways within that electromagnetic field by the selection of a number of predetermined central landmarks. A steerable catheter with a sensor tip to locate the position of the catheter within 3D space is then used to navigate biopsy tools toward the lesion without the need for visualization of the pathway [12]. The diagnostic yield reported using ENB is 73% [13].

An emerging additional technique is bronchoscopic transparenchymal nodule access. Using this technique, a pathway to the lesion is created through the parenchyma. There is still a very small number of studies available. A diagnostic yield of up to 85% has been reported [12,14,15]. However, real-world data using navigation-guided bronchoscopy showed a significantly lower diagnostic yield in comparison with the early studies [13].

Robotic-assisted bronchoscopy is an innovation that aims to improve diagnostic yield in comparison with other navigation techniques. Although this technique might be a step forward toward the improvement of diagnostic yield, its real additional value needs to be determined [16]. All these technological advances could allow more precise endobronchial treatment of possible malignant lesions.

### 2.2. Inhalation Therapies

The large surface area of the lungs and alveoli is suitable for drug deposition, making it a possible application route [17]. Despite this potential, inhalation therapy does not show promising results, which is likely due to difficulties in the preparation and delivery of the agent. Production of this type of drug differs based on the chemical properties, such as hydrophilic and hydrophobic molecules. In addition, the particles must be small (1–3 µm) to allow deposition in the alveoli [18,19]. Because of these limitations, the effect of inhalation therapies is suspected to be restricted to the upper airways. Human trials testing the feasibility, efficacy and safety are limited. The development of inhalation therapies does not evolve beyond phase II clinical trials at best [20]. To simplify the intake of inhalation drugs, inhalers and nebulizers have been developed [17]. Despite the effort of creating these devices, the lungs receive only a small percentage of the amount administered. So far, inhalation therapies are therefore less applicable for treating lung cancers than initially expected.

### 2.3. Navigation Therapies

#### 2.3.1. Direct Tumor Injection

During direct tumor injection, the chemotherapeutic agent is delivered through a needle into the tumor. Transbronchial aspiration needles are commonly used for this treatment, although more suitable needles are under investigation. The agent is administered in a 1:1 ratio to the volume of the tumor [21].

The effect of cisplatin on advanced NSCLC and SCLC using a transbronchial aspiration needle was studied by Mehta et al. [22,23]. A complete response (CR) or partial response (PR) was seen in 69% of the population. Responders showed an overall survival (OS) of 10 months and progression-free survival (PFS) of seven months, while the values for non-responders were respectively six and three months. The reported adverse events (AEs) were transient nausea and broncho-mediastinal fistula. In 2019, a prospective multicenter trial with a novel transbronchial microneedle injection catheter was tested for NSCLC tumors [24]. This needle can inject the tumor perpendicularly, which increases the accessibility of tumors and allows a higher percentage of the airway wall to be affected per injection. After 12 weeks, no differences were found in tumor volume and health-related quality of life (QOL), suggesting that it is a safe and feasible option. The results reported above are the effect of direct tumor injection combined with chemotherapy or radiotherapy. The contribution of the co-administered therapy cannot be excluded, which results in overestimating the impact of direct tumor injection. Further trials are therefore needed.

#### 2.3.2. Brachytherapy

In the early 1900s, physicians struggled to pass radiation through healthy tissue to reach the tumor [25]. To solve this problem, the radioactive source was placed into the patient’s body. This technique is known as brachytherapy, planting seeds or internal radiotherapy. Depending on the position of the source, various designations are given to internal radiotherapies, such as interstitial brachytherapy and intracavity brachytherapy. In interstitial brachytherapy, the radioisotope is placed directly into the tumor, which differs from intracavity brachytherapy where it is placed into a cavity located nearby the malignancy. Depending on the duration of implantation and radiation dose, there are three different types of internal radiotherapy: low-dose rate (LDR) implants, high-dose-rate (HDR) implants and permanent implants [25]. The most used clinical radioisotopes are ^125^I and ^103^Pb.

Chen E. et al. and Li et al. studied brachytherapy without concomitant therapies and reported an efficiency of 90% and a local control (LC) rate of 99% when treating NSCLC. [26,27]. An increase in LC was related to a prescription dose higher than 120 Gy. A total of 23% of the cases led to complete remission, while 66% led to partial remission. In the more advanced stages of NSCLC, the OS rates after one and two years were 82.8% and 37.1% [27]. It is essential to mention that these survival rates are speculated to be higher in early-stage NSCLC, in which there is no metastatic disease. Several clinical studies investigated the effect of brachytherapy as an adjuvant therapy combined with other modalities such as chemotherapy, radiofrequency ablation and epidermal growth factor receptor tyrosine kinase inhibitors [28,29,30,31,32]. The addition of brachytherapy to the other modalities showed an increased effect, OS, PFS and QOL. The most common AEs related to internal radiotherapy are pneumothorax (19%), bleeding (14%) and radiation pneumonitis (6%) [26,27].

An important limitation of brachytherapy is that implantation quality affects the treatment outcome [33]. Internal radiotherapy requires highly trained personnel, a lot of equipment and planning, making it an expensive and labor-intensive treatment. Most clinicians therefore tend to use alternative techniques that are easier, cheaper and less time-consuming [33,34].

#### 2.3.3. Radiofrequency Ablation (RFA)

RFA is based on creating an alternating radiofrequency current between the electrode placed into the lesion and the dispersive patch placed on the patient’s body. Endobronchial RFA requires a bronchoscope with an RFA probe positioned at the end. The RF current generates an oscillating electric field, causing the surrounding ions to vibrate. The kinetic energy produced by the oscillating ions is converted into thermal energy or heat. When the temperature in the tumor rises above 60 °C, protein denaturation and coagulation is induced, causing cell death. One of the limitations of RFA is the spread of heat to neighboring blood vessels, also known as the “heat sink” effect, causing a decrease in efficacy. This effect only applies to vessels located within a range of 1 cm, and with a diameter of ≥3 mm.

The results of three patients treated with fluoroscopic guidance were described in a case series [35]. All patients showed a response to the treatment, 66.6% partial and 33.3% complete. After one year, two out of three patients showed a PFS. No serious AEs were observed in these cases. In 2022, a pilot study of five patients investigated EBUS-RFA for treating centrally located early-stage NSCLC [36]. The observed ablation diameter ranged from 1.8–2.6 cm. Adverse events did not occur immediately after the procedure. Further research is required to investigate its safety and efficacy in humans. Regarding the limited data, endobronchial RFA needs further exploration, and it is too soon to name it a promising technique for managing smaller NSCLCs.

#### 2.3.4. Microwave Ablation (MWA)

Like transbronchial RFA, transbronchial microwave ablation involves placing a bronchoscope with probe into the tumor, emitting electromagnetic waves into the surrounding tissue. These waves cause the water molecules to oscillate, resulting in a temperature rise (>60 °C), thermocoagulation and thus, cell death of the surrounding tissue.

Kashiwabara et al. reported an OS of six months for patients with advanced NSCLC or lung metastases [37]. A total of 4 of the 28 participants showed complications such as endobronchial bleeding and re-occlusion within 24 h. Transbronchial MWA showed a technical success rate of 100% when treating lung nodules, early-stage NSCLC and early-stage SCLC patients [38,39]. A pilot study by Xie et al. (2022) showed a complete response (CR) of 78.6% and a two-year LC rate of 71.4% [39]. The number of AEs observed during these studies is limited [37,39]. The most common ones are pleuritic chest pain (13.3%), pneumothorax (6.67%), hemoptysis (3.33%) and infected effusion (3.33%) [38]. The observed median PFS was 33 months. Additional research is required to investigate the safety and effectiveness of this technique in larger human populations.

#### 2.3.5. Cryoablation (CRA)

During CRA, argon gas migrates through the centrally located high-pressure chamber toward the tip of the cryoprobe, passing into an outer low-pressure chamber. The transition from high to low pressure results in a temperature decrease, which is known as the Joules–Thompson effect [40]. This drop in temperature (−30 °C to −70 °C) will lead to the formation of intra- and extracellular ice crystals. An alternating cycle of freezing and thawing creates direct and indirect cell damage. Direct effects are caused by intra- and extracellular ice formation, disrupting cellular processes. Indirect cellular damage is the result of vasoconstriction and edema.

A systematic review by Lee et al. included 16 studies representing a total of 2355 patients, mostly presenting with advanced lung or bronchial tumors [41]. The observed CR rates ranged from 61–100%. The two-year survival rate was approximately 60%, with an average survival of 8.2 months.

In 10 of the included studies, complications were seen in 11.1% of the cases, including hemorrhage, mediastinal emphysema, atrial fibrillation and dyspnea. A total of 5 of the 16 studies reported a mortality rate of 7.1% within a month of the operation. By analogy with RFA, the results of cryoablation are also impacted by the heat sink effect [41].

#### 2.3.6. Photodynamic Therapy (PDT)

PDT is a form of phototherapy in which light is used to treat a patient. During PDT, a photosensitizing agent is administered to actively or passively accumulate into the tumor. Photosensitizers are known to produce reactive oxygen, after excitation by near-infrared light, due to their reaction with molecular oxygen [42]. Cell death is initiated directly (apoptosis, autophagy and necrosis) or indirectly (inflammatory response) by reactive oxygen.

A prospective phase II study evaluated 54 patients with centrally located early-stage lung cancers treated using a first-generation photosensitizer (Photofrin II) [43]. Despite the CR rate of 84.8%, tumor size appears to be the prognostic factor, as suggested by the better response for tumors ≤ 1.0 cm. The observed AEs were pulmonary toxicity (7.7%), allergic reaction (7.7%), the elevation of ALT (1.9%) and sunburn (1.9%). Both a clinical trial by Kato et al. and Dhillon et al. confirmed that PDT has a response rate ranging from 80 to 90% [44,45]. In 2013, a pilot study was published using Radachlorin^®^ PDT in advanced NSCLC [46]. Radachlorin^®^ is a second-generation photosensitizer with better pharmacokinetics. Complete or partial response occurred in 20% and 70% of cases, respectively. A reduction in symptoms caused by airway obstructions resulting from PDT treatment has been seen [46,47]. The observed one-year survival rate was 70%. A prospective study reported a correlation between disease stage and median survival rates [48]. The median OS was seven months, while for the specific stages, the results were, respectively: not reached (I), 22.5 months (II), 5.7 months (IIIA), 5.5 months (IIIB) and 5.0 months (IV).

PDT has some limitations, making it unsuitable for deep-seated tumors and metastases [49]. The poor penetration capacity of near-infrared light causes a decreased effect on deep-seated tumors. As mentioned, photosensitizers must be incorporated into the lesion to cause an impact, which is only possible in localized solid tumors. This characteristic makes PDT unsuitable for treating metastases. During the drug–light interval, patients must pay attention to light exposure, which causes discomfort in their daily routines. The drug–light interval is the time between the drug’s administration and the light application. Depending on which photosensitizer is used, the duration of this interval can range from a few hours to several days.

## 3. Endovascular Administration

### 3.1. Bronchial Circulation

#### Bronchial Artery Infusion (BAI)

Oxygenated blood reaches the lungs through the bronchial arteries, which arise from the descending thoracic aorta. BAI is a non-invasive treatment to administer chemotherapeutic agents directly into the bronchial artery. Using the Seldinger technique, a catheter is introduced into the femoral artery. Once positioned correctly, the therapeutic agent is injected through the lumen of the catheter.

A prospective study by Nakanishi et al. evaluated BAI, using docetaxel and cisplatin without administering systemic chemotherapy, as a treatment modality for stage III and IV (M1a) NSCLC [50]. An overall response rate of 52% was found, mainly including a PR. After 18.5 months, the median OS and PFS were 6.5 and 17.4 months. The one- and two-year survival rates were 81% and 32%. In 2017, a randomized controlled trial was published including 46 patients, of which 24 were treated with BAI only and 22 received BAI combined with icotinib hydrochloride [51]. The total number of BAI regimens was 3.71 ± 1.46, causing a response rate of 29%. After one year, the QOL improved by 58%. Zhu et al.’s single-arm retrospective study tested BAI as a neoadjuvant therapy in 36 patients with unresectable stage III squamous cell lung cancer [52]. They reported a response rate of 72.2% and a downstaging of 22.2%. The OS and PFS were, respectively, 25 months and 14 months. The one- and two-year survival rates were 75.4% and 52.1%, respectively.

Mild toxicity and AEs have been observed during BAI. The most common side effects are fatigue (36%), leukopenia (28%), neutropenia (28%), loss of appetite (28%), chest pain (24%), cough (20%), nausea (16%), diarrhea (12%) and alopecia (9%) [50]. The disadvantages of BAI are the requirements of many materials and expertise, possible complications of collateral damage to other structures perfused by the vessel and systemic effects.

### 3.2. Pulmonary Circulation

#### 3.2.1. Selective Pulmonary Artery Perfusion with Blood Flow Occlusion (SPAP-BFO)

Selective pulmonary artery perfusion (SPAP) is a non-invasive treatment modality for the regional administration of chemotherapeutics. Insertion of a balloon catheter is performed by puncturing the femoral vein. After insertion, the catheter is positioned into the left or right pulmonary artery. Once set correctly, the balloon is insufflated until a blood flow occlusion (BFO). Chemotherapy is given through the lumen of the balloon catheter. The distal flow reduces due to the obstruction, giving the therapeutic agent more time to be absorbed by the lungs.

To date, no human trials have been published studying SPAP followed by BFO as a treatment modality for pulmonary malignancies. However, a limited number of preclinical studies have been conducted using a porcine model. In a study by van Putte et al. (2008), a significantly larger lung concentration AUC was seen when applying 10 min of SPAP compared to 2 min [53]. Grootenboers et al. observed higher AUC of lung tissue when extending the BFO time [54]. The highest AUC values were obtained when applying 1.25 g/m² gemcitabine for two minutes combined with 30 min BFO. In 2009, van Putte et al. proved that SPAP-BFO of gemcitabine and/or cisplatin resulted in significantly increased drug exposure in the lungs compared to intravenous application [55]. Sparks et al. observed significantly increased concentrations in hilar and mediastinal lymph nodes using 30 min SPAP-BFO compared to intravenous administration [56].

SPAP-BFO is considered to be an effective technique for the local administration of chemotherapy, reducing systemic AEs. Although the results mentioned above are promising, these porcine models lack data surrounding the technical feasibility in humans. During the aforementioned experiments, the catheter was placed in the pulmonary artery directly or via the left internal jugular vein, in contrast to human application where it is introduced via the femoral vein. Therefore, clinical trials are required to give more clarity surrounding the feasibility, safety and effectiveness of these techniques in humans.

#### 3.2.2. Chemoembolization

A mix of a chemotherapeutic and an embolic agent is injected into the arteries to induce embolization. Cytostatic agents applied during this technique are mitomycin C, cisplatin, gemcitabine and doxorubicin. This embolism will block the tumor’s blood supply and trap the drug inside. The therapy is limited to the tumor-feeding arteries by specific targeting using image-guided catheterization. Different embolic agents with specific embolization mechanisms have been developed, including lipiodol, drug-eluting beads and degradable starch microspheres [57]. Embolization caused by lipiodol or degradable starch is transient, with occlusion half times of 4–12 weeks and 90–120 min. The use of drug-eluting beads has led to irreversible embolization of the artery.

A study by Jin et al. investigated the clinical efficacy, QOL and AEs in lung cancer patients treated with transcatheter arterial chemoembolization [58]. Two hundred sixty-four patients were treated with chemoembolization, including stages IIA (52.3%), IIIB (26.9%) and IV (20.8%). Chemoembolization showed a total effectiveness of 80.3%, consisting of 48.9% CR and 31.4% PR. A significant increase in the Karnofsky Performance Status (KPS) score was observed over time, indicating increased patient health. A phase I study, including 10 patients with unresectable lung metastases, revealed a technical success rate of 100% [59]. According to the Response Evaluation Criteria in Solid Tumors, the response rate was 10%, but when using the PET response criteria it was 40%. Vogl et al. reported a mean survival of 17 months and a PR rate of 30.7% [60].

The observed AEs were nausea and vomiting (4.9%), dizziness (4.2%), abdominal pain (2.7%), abdominal distension (1.9%), allergy (1.1%) and renal function injury (1.14%). A meta-analysis, including four randomized clinical trials, revealed that dexamethasone administration significantly reduces the incidence rate of chemoembolization-related AEs [61].

## 4. Transthoracic Administration

### 4.1. Isolated Lung Perfusion (IluP)

Isolated lung perfusion (ILuP) requires cannulation of the pulmonary artery and veins to connect them to an extracorporeal perfusion system. This system consists of a perfusate reservoir, a pump and a deoxygenator. A thoracotomy is performed to access the artery and veins. After installation of the perfusion system, the chemotherapeutic agent is injected into the enclosed system to perfuse the lungs.

Published studies investigating ILuP are restricted to lung metastases only. A clinical phase I trial by den Hengst et al. (2010) showed a five-year OS rate of 54.8% with an overall median survival time of 84 months [62]. The five-year disease-free survival (DFS) rate and median survival time were 27.5% and 19 months. During a clinical phase II study by den Hengst et al. (2014), the median time to local pulmonary progression was not reached within 24 months [63]. They reported a three-year OS and DFS of 57% and 36%, respectively. Beckers et al. observed an incidence of local recurrence of 43%, with a median progression and survival time of 14 and 54 months [64]. Several clinical studies investigated the pharmacokinetics of isolated lung perfusion [65,66,67]. When using doxorubicin, the maximum tolerated dose was 40 mg/m². At 37 °C, the maximum tolerated dose of melphalan is 60 mg.

After 12 months, patients treated with ILuP showed a decrease in lung function (i.e., vital capacity, total lung capacity, forced expiratory volume in one second and diffusion capacity of the alveolocapillary membrane) ranging from 7.9 to 11.3% [63]. The results of ILuP demonstrated that loco-regional techniques could be administered without systemic exposure. Because of the invasive nature of this technique, other less invasive regional techniques are currently being evaluated, including SPAP.

### 4.2. Suffusion by Video-Assisted Thoracoscopic Surgery (VATS)

Suffusion involves regulating the blood supply and drainage by blocking the pulmonary artery and veins. Occlusion of the pulmonary artery is performed using transfemoral insertion of a balloon catheter. Snaring of the veins is performed by video-assisted thoracoscopic surgery (VATS). The chemotherapeutic agent is injected via the lumen of the catheter. Therefore, it is a hybrid technique that combines a thoracoscopic and endovascular approach.

Data regarding this technique are limited. A preclinical pilot study using four dogs showed that 75% of the agent remained within the lung [68]. Suffusion has been tested in a clinical phase I study, including four patients with stage IV oligometastasis [69]. They reached a technical success rate of 100%. Compared to the systemic concentration of 236 ng/mL, a high concentration of cisplatin was found within the pulmonary circulation, at 1124 ng/mL. Patients showed no differences in their pulmonary function tests and scans before and after the procedure. Further studies are required to clarify the safety and efficacy of suffusion.

Single-lung ventilation will be used during this technique, for which a sufficient pulmonary reserve is required. This makes suffusion less suitable for patients with significant pulmonary comorbidity and/or limited pulmonary function.

### 4.3. CT-Guided RFA

An RFA probe will be installed under Computed Tomography (CT) guidance into the tumor. The principle of percutaneous CT-guided RFA is similar to endobronchial RFA. The ablation zone, where cell death occurs, is formed spherically around the probe. A single probe can create a limited ablation zone of 5 cm [70]. A correlation was found between tumor recurrence and the difference between the diameter of the ablation zone and the tumor index after two years [71]. Therefore, it is important to induce an ablation zone larger than the tumor. A study by Zhang et al. showed that the recurrence rate reduces by 52% for each centimeter increase in the ablation zone diameter compared to the tumor index [72]. These properties make RFA only suitable for treating small tumors, such as stage I NSCLC.

Percutaneous RFA has been widely studied as a treatment for early-stage NSCLC tumors. A meta-analysis by Li et al. revealed a technical success rate between 79.50 and 100% and a local progression rate of 26% [73]. RFA showed a local progression rate of 19.8–26.0% [73,74]. A meta-analysis by Zhang et al. reported LC rates at one and five years of 75% and 41%, respectively [72]. These data are in line with the one- and three-year LC rates reported by Palussière et al. [75]. Three studies reported OS rates after one and three years of 84.3–91.7% and 48.0–62.4% [72,74,75]. The median survival time, reported by Yuan Z et al., was 30.9 months [76]. A study by Chen S. et al. reported a local control rate of 73.3% at three months, 66.7% at six months and 40.0% at nine months [28]. The most common AEs reported for RFA are pneumothorax (27.2%), pleural effusion (4.1%) and hemoptysis (2.2%) [72].

CT-guided RFA is a safe and feasible technique for treating peripheral early-stage lung cancers. A limitation of CT-guided ablation techniques in general is their limited accessibility for treating centrally positioned tumors.

### 4.4. CT-Guided MWA

During CT-guided percutaneous MWA, the probes will be installed as in percutaneous RFA. The ablation zone of one MWA probe is 5–6 cm, but this can be enlarged by using multiple probes. In addition to the probe, an antenna is placed near the targeted lesion to monitor intratumoral temperatures. Compared to other ablation techniques, MWA has some advantages: longer ablation times, uniform and continuous heath delivery, lower sensitivity to the heat sink effect and an extendable ablation volume.

A meta-analysis by Yuan et al. showed a complete ablation rate of 81.1% and a median survival time of 25.6 months. The OS rates of WMA at one, two, three, four and five years are 82.5%, 52.6%, 35.7%, 29.6% and 16.6%, respectively [74]. Four recently published randomized clinical trials by Xu et al. evaluated local progression, survival prediction and risk prediction of pleural effusion or pneumothorax [77,78,79,80]. The reported OS is 15–17 months, with a local progression rate of 23.0%. The OS depends on the tumor diameter, extrapulmonary metastases, tumor stage, tumor type and post-MWA tyrosine kinase inhibitors. The reported predictors for increased local progression are tumor subtypes (others > squamous cell carcinoma > adenocarcinoma), vessels ≥ 3 mm in direct contact with the tumor, tumor diameter ≥ 3 cm and localization in the upper lobe. The PFS for NSCLC treated with MWA is 12.5 months. The main AEs associated with MWA are pneumothorax (28.9%) and pleural effusion (5.9%) [77,78,79]. A clinical trial phase III study revealed that the combination of MWA and chemo could prolong the OS and PFS in patients with advanced NSCLC, compared to chemotherapy alone [81].

MWA can be used as a primary treatment modality for stage I and II NSCLC. In contrast to RFA, MWA can be used for more extensive and advanced tumors due to the extensibility of the ablation volume. Combining MWA with chemotherapy or radiotherapy is a possible treatment modality for stage III and IV NSCLC.

### 4.5. CT-Guided CRA

The CRA probes are placed directly into the patient’s chest, till positioned correctly. This approach makes percutaneous CRA more invasive compared to endobronchial application. The ablation zone area of a single cryoprobe has a diameter of 2–3 cm and a length of 4 cm. Similar to MWA, the use of multiple probes allows an enlargement of the ablation zone.

De Baere et al. investigated CRA to treat lung metastases in 40 patients [82]. They obtained local tumor control rates of 96.6% after six months and 94.2% after 12 months. The one-year OS rate was 97.5%, while no differences in QOL were found. A meta-analysis observed the OS rate after one, two and three years being 86.5%, 73.5% and 71.2%, respectively [74]. The authors also reported a local progression rate of 10.9%. A recent randomized controlled trial performed by Yang et al. reported the response of 41 stage III and IV NSCLC patients compared to co-ablation therapy [83]. Co-ablation therapy exists as a combination of CRA and thermal ablation using a single probe. No differences were found between the groups concerning disease control (95%) and response rate (87.5%), suggesting that CRA and co-ablation are equally effective. Additionally, a correlation was found between DCR and the diameter of the lesion.

The observed complications caused by CRA are similar to those of the ablation techniques. They include pneumothorax, pleural effusion and hemoptysis. Comparable to RFA, the results of cryoablation suffer from the heat sink effect.

For the routine clinical implementation of the discussed techniques, many more studies are needed. This is due to the limited number of clinical trials to guarantee their effectiveness and safety. Pending these additional results, the current clinical standard for early-stage cancer remains minimally invasive surgery, such as VATS. In case of advanced cancers chemo-, radio- and immunotherapy can be used.

## 5. Conclusions

There has been a growing interest in the development and improvement of loco-regional administration techniques for treating lung cancer. These novel techniques have shown improved local disease control and effect. Endobronchial administration techniques are applicable in the treatment of a specific lung area or solid tumor with a bounded region. Although data are still limited, these endobronchial ablation techniques show similar results compared to the percutaneous approach. The advantage of endobronchial ablation is the reduction in complications and limited discomfort for the patient. This is related to the less invasive nature of the procedure. However, further investigation is needed to assess the safety and effectiveness of these techniques in larger randomized controlled trials. Endovascular techniques are effective and safe for treating larger areas of the lung or lung metastases. This type of procedure can be used to treat affected regions which are hard to reach surgically or to shrink tumors to allow resection or thoracotomy. Transthoracic administration techniques are more invasive approaches to treat lung malignancies. If a large area of one of both lungs is affected, ILuP and lung suffusion can be used. In the case of well-defined solid tumors, CT-guided percutaneous ablation techniques are applicable. The interest in optimizing transthoracic techniques is highly dependent on future results of the bronchial and endovascular procedures. If these techniques achieve at least equal results, the interest in transthoracic techniques will decrease due to their invasiveness and severe complications. Further development of loco-regional administration techniques will enhance the success rate for treating lung cancers, and reduce both AEs and the impact of the treatment on patients.

## Figures and Tables

**Figure 1 cancers-15-02244-f001:**
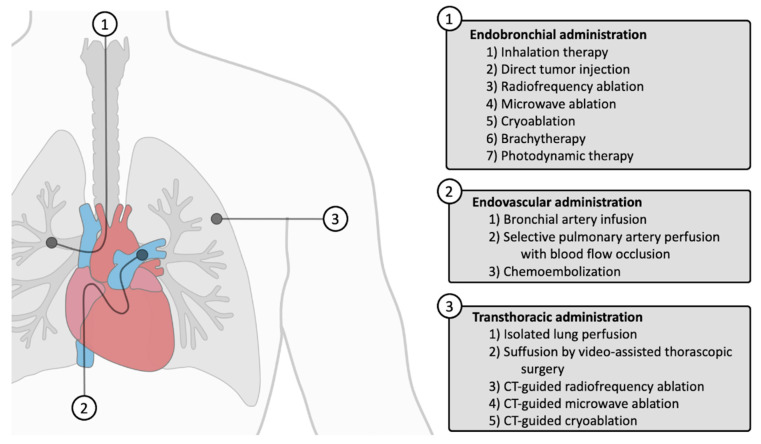
Overview of the loco-regional techniques based on their route of administration.

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
