# Peer review of "Innovative Invasive Loco-Regional Techniques for the Treatment of Lung Cancer"

_cancers, 2023, doi:10.3390/cancers15082244_

Round 1

Reviewer 1 Report

I have read the article with anticipation, especially the  chapter about inhalations and brachytherapy.

However I have some remarks:

1.       I would argue with the statement line  42:  

Unfortunately, most cancers get diagnosed in more advanced disease stages, including stages IIb, III and IV, for which surgical resection is usually not recommended.

Please provide publication why stage IIb is not recommended for surgery.

2.       Chemoembolization - Authors do not state what cytostatic agents  were given. It might be interesting to add.

3.       KPS might not be understood for common reader ( please explain eng. Karnofsky score ? )

4.       The authors don’t mention PET CT as a preferable diagnostic tool to exclude metastatic changes and therefore avoid ineffective surgery  or  only local treatment.

5.        In the era of immunotherapy  and targeted therapy adjuvant treatment plays a crucial role. Adjuvant methods , especially targeted treatment based on ADURA trail should be further discussed. It is mention in the table but has no context in the article

Author Response

I have read the article with anticipation, especially the chapter about inhalation and brachytherapy. However, I have some remarks:

Comment 1: I would argue with the statement line 42: “Unfortunately, most cancers get diagnosed in more advanced disease stages, including stages IIb, III and IV, for which surgical resection is usually not recommended.” Please provide publication why stage IIb is not recommended for surgery.

Answer 1: We understand your comment about the statement mentioned in line 42. Generally, stage IIb is resectable and can be treated with surgery. We mentioned stage IIb because some of the patients might suffer from comorbidities making them no ideal candidates to undergo surgery. To avoid the misconception surrounding this statement, we have no longer mentioned stage IIb in this statement.

We changed the statement to: “Unfortunately, most cancers get diagnosed in more advanced disease stages, including stages III and IV, for which surgical resection is usually not recommended.” (Line: 57)

Comment 2: Chemoembolization – Authors do not state what cytostatic agents were given. It might be interesting to add.

Answer 2: We agree that it might be more interesting to add more information about the cytostatic agent to the text. Boas F.E. et al (59) and Vogl T.J. (60) et al used mitomycin C as cytostatic agent, during their study. Other cytostatic agents used for this technique are cisplatin, gemcitabine and doxorubicin (60).

We added the following text: “Cytostatic agents applied during this technique are mitomycin C, cisplatin, gemcitabine and doxorubicin.” (Line 318-319)

Comment 3: KPS might not be understood for common reader (please explain eng. Karnofsky score?)

Answer 3: We agree with this comment and believe that KPS needs some additional clarification for the common readers. KPS stands for Karnofsky Performance Status, which is a tool to assess functional impairment. The lower this score, the worse change of survival. The score ranges from 0 (= dead) to 100 (= normal no complaints).

We made the following changes: “A significant increase in Karnofsky Performance Status (KPS) score was observed over time, indicating increased patient health.” (Line 330-331)

Comment 4: The authors don’t mention PET CT as a preferable diagnostic tool to exclude metastatic changes and therefore avoid ineffective surgery or only local treatment.

Answer 4: We would like to thank the reviewer for this remark, we agree that PET-CT as diagnostic tool deserves a place within this review. We have added a paragraph regarding the position of PET-CT within the diagnostic field to our introduction.

We added the following: “Positron emission tomography and computed tomography (PET-CT) is an essential diagnostic tool in the management of lung cancers. It combines functional and anatomical information, allowing precise disease staging and thus better decision making in treatment options.” (Line 48-51)

Comment 5: In the era of immunotherapy and targeted adjuvant treatment plays a crucial role. Adjuvant methods, especially targeted treatment based on ADURA trail should be further discussed. It is mentioned in the table but has no context in the article.

Answer 5: In order to give more clarity about the adjuvant targeted treatments. We inserted the following lines: “In addition to these standard treatments, other options include targeted therapy and loco-regional techniques. Targeted therapy blocks the expression of specific molecules associated with cancer growth and progression. It is mainly used to treat NSCLC with mutations in EGFR, ALK, ROS1, BRAF, MET, RET and NTRK genes.” (Lines 61-65)

Reviewer 2 Report

I would like to congratulate the authors of an interesting article entitled "Innovative invasive loco-regional techniques for the treatment of lung cancer".  The article is written in an accessible way, in good quality English. The refer to the current literature and thoroughly analyze the topic of innovative, minimally invasive methods of lung cancer treatment.

I have only one comment to the article. All of the methods described are now considered to be, at most, alternatives to the treatments currently recommended by lung cancer societies. In order to avoid scientific, clinical and legal accusations of promoting treatment methods with unproven effectiveness and an unknown safety profile, I would suggest adding a short section (just before conclusions?) consisting of a few sentences in which the authors would collectively define the limitations of the above methods (efficacy, safety, lack of clinical trials comparing to currently used methods) , and the like) and would emphasize the role of currently used methods (minimally invasive surgical treatment in early-stage lung cancer, chemo-, radio-, immunotherapy in advanced cancer).

Once again, I would like to congratulate you on an interesting article.

Author Response

I would like to congratulate the authors of an interesting article entitled "innovative invasive loco-regional techniques for the treatment of lung cancer". The article is written in an accessible way, in good quality English. They refer to the current literature and thoroughly analyze the topic of innovative, minimally invasive methods of lung cancer treatment.

I have only one comment to the article. All of the methods described are now considered to be, at most, alternatives to the treatments currently recommended by lung cancer societies. In order to avoid scientific, clinical and legal accusations of promoting treatment methods with unproven effectiveness and an unknown safety profile. I would suggest adding a short section (just before conclusions?) consisting of a few sentences in which the authors would collectively define the limitations of the above methods (efficacy, safety, lack of clinical trials comparing to the currently used methods) and would emphasize the role of currently used methods (minimally invasive surgical treatment in early-stage lung cancer, chemo-, radio-, immunotherapy in advanced cancer).

Once again, I would like to congratulate you on an interesting article.

Answer comment:  We would like to thank the reviewer for this comment; we agree that it seems helpful to add to our text.

We added the following paragraph to comply with your comment: “For the routine clinical implementation of the discussed techniques, a lot more studies are needed. This is due to the limited number of clinical trials to guarantee their effectiveness and safety. Pending these additional results, current clinical standards for early-stage cancer remain minimally invasive surgery, such as VATS. In case of advanced cancers chemo-, radio- and immunotherapy can be used.”. (Line 454-459).